# Neuronal expression in Drosophila of an evolutionarily conserved metallophosphodiesterase reveals pleiotropic roles in longevity and odorant response

**Kriti Gupta**[ID][◉], **Sveta Chakrabarti**[ID][◉], **Vishnu Janardan**[◉], **Nishita Gogia**, **Sanghita Banerjee, Swarna Srinivas, Deepthi Mahishi, Sandhya S. Visweswariah**[ID]*

Department of Developmental Biology and Genetics, Indian Institute of Science, Bengaluru, India

◉ These authors contributed equally to this work.
* sandhya@iisc.ac.in

**Data Availability Statement:** All original data from the RNA-seq has been deposited in ArrayExpress

## Abstract

Evolutionarily conserved genes often play critical roles in organismal physiology. Here, we describe multiple roles of a previously uncharacterized Class III metallophosphodiesterase in *Drosophila*, an ortholog of the MPPED1 and MPPED2 proteins expressed in the mammalian brain. dMpped, the product of *CG16717*, hydrolyzed phosphodiester substrates including cAMP and cGMP in a metal-dependent manner. *dMpped* is expressed during development and in the adult fly. RNA-seq analysis of *dMpped*^KO flies revealed misregulation of innate immune pathways. *dMpped*^KO flies showed a reduced lifespan, which could be restored in *Dredd* hypomorphs, indicating that excessive production of antimicrobial peptides contributed to reduced longevity. Elevated levels of cAMP and cGMP in the brain of *dMpped*^KO flies was restored on neuronal expression of dMpped, with a concomitant reduction in levels of antimicrobial peptides and restoration of normal life span. We observed that *dMpped* is expressed in the antennal lobe in the fly brain. *dMpped*^KO flies showed defective specific attractant perception and desiccation sensitivity, correlated with the overexpression of *Obp28* and *Obp59* in knock-out flies. Importantly, neuronal expression of mammalian MPPED2 restored lifespan in *dMpped*^KO flies. This is the first description of the pleiotropic roles of an evolutionarily conserved metallophosphodiesterase that may moonlight in diverse signaling pathways in an organism.

## Author summary

The MPPED2 gene maps to a human genetic locus associated with the WAGR syndrome, manifesting as Wilms Tumor, genitourinary abnormalities, and mental retardation. The mammalian protein is expressed in the brain, but the function of this protein in neurons is unknown. Here we have used Drosophila to identify various functions of the fly ortholog of MPPED2. We find that this gene product is expressed in a variety of tissues and can cleave phosphodiester bonds present in cyclic nucleotides. The protein is important for

with accession E-MTAB-9081 (https://www.ebi.ac.uk/biostudies/arrayexpress/studies/E-MTAB-9081).

**Funding:** Financial Disclosure Statement Funding from the DBT-IISc Partnership Program Phase-II BT/PR27952/INF/22/212/2018/21.01.2019 is acknowledged (https://dbtindia.gov.in). SSV is a JC Bose National Fellow (SB/S2/JCB-18/2013; https://www.serbonline.in/SERB/jcbose_fellowship) and a Margdarshi Fellow supported by the Wellcome Trust DBT India Alliance (IA/M/16/1/502606; https://www.indiaalliance.org/about-us/about-ia). The funders had no role in study design, data collection and analysis, decision to publish, or preparation of the manuscript.

**Competing interests:** The authors have declared that no competing interests exist

optimum life span in the fly, mediated by regulating the expression of immune genes. Longevity could be rescued by neuronal expression of the mammalian ortholog. The gene product is expressed in neurons in the antennal lobe of the fly and modulates responsiveness to odorants. Therefore, this evolutionarily conserved protein has multiple roles in the physiology of an organism, either by interacting with other proteins, or cleaving natural phosphodiester bonds, and further studies in mammals are warranted.

## Introduction

Evolutionarily conserved proteins usually play a role in fundamental biological processes in an organism [1,2]. Sequence conservation at the level of amino acids, especially at the catalytic site of an enzyme, implies strong structural conservation and perhaps common activities in organisms separated by large tracts of time. If a gene linked to a genetic disease in humans has a counterpart in lower organisms more amenable to genetic manipulation, important insights into the gene's function can be gained by either deletion or overexpression of the ortholog in a simpler organism. Such studies pave the way toward defining approaches that can be utilized in mammalian model systems later.

WAGR syndrome (Wilms' tumor, aniridia, genitourinary anomalies, mental retardation) [3] is associated with interstitial deletions in a region in chromosome 11p13 [4,5]. This locus contains several genes such as *WT1*, *BDNF* [6], *PAX6* [7] (all important during development), *FSHB* [8], *RCN1*, and *MPPED2* (*239FB*) [9,10]. *MPPED2* mRNA is predominantly expressed in the fetal brain [11], and our earlier biochemical characterization revealed that this protein belonged to the large family of metallophosphoesterases with promiscuous utilization of substrates [9,12]. We could identify orthologs in all mammals and other vertebrates, including *Caenorhabditis elegans* and *Drosophila melanogaster* [9,12]. Indeed, a closely related structural ortholog is found in *Mycobacterium tuberculosis* [13]. There are two variants of the MPPED family in higher organisms, MPPED1, and MPPED2 (239AB) [14], and these two mammalian orthologs show more than 80% similarity at the amino acid level, are expressed in the brain and share similar biochemical properties [9]. Recent studies have shown that MPPED2 is one of the risk loci for migraine [15], and is associated with altered systemic inflammation and increased organ dysfunction in trauma patients [16]. Further, MPPED2 is downregulated in glioblastoma [17] and thyroid neoplasia [18], acts as a tumor suppressor in breast cancer [19] and oral squamous cell carcinoma [20], and the MPPED2 gene is differentially methylated in colorectal cancer [21] and in individuals with gender incongruence [22]. These reports suggest diverse roles for MPPED2 in multiple tissue types.

The metallophosphoesterases represent a large and diverse group of proteins with similar structural folds that harbor two essential metal ions at the catalytic site [12]. Rat Mpped1 and Mpped2 hydrolyze phosphodiester bonds [9]. The substrates for many of these enzymes remain unknown, and while Mpped1 and Mpped2 can hydrolyze 3',5'-cAMP, the product of this reaction is 3'AMP [9,23], in contrast to well-characterized mammalian cyclic nucleotide phosphodiesterases that produce 5' AMP as the hydrolysis product [24]. Interestingly, 2'3'-cAMP is the preferred substrate for this group of enzymes, forming 3' AMP and 2' AMP as products [25], but the biological relevance of this reaction is unknown. MPPED1 and MPPED2 are also able to hydrolyze several colorigenic substrates, and such assays revealed that these enzymes could not hydrolyze monoesters but only diester-containing molecules such as bis-p-nitrophenyl phosphate (bis-pNPP), p-nitrophenyl phenylphosphonate (p-NPP) and TMP p-nitrophenyl ester (TMPP) [9].

*D. melanogaster* harbors a single ortholog of MPPED1/MPPED2, which we have named dMpped [9]. In FlyBase, this gene is annotated as *Fbgn0036028* or *CG16717*. There is no information available on the role of this gene product to date. To decipher the role of *dMpped* in the fly, we have biochemically characterized this protein and found that it is a phosphodiesterase that can hydrolyze cAMP and cGMP. The major sites of expression of *dMpped* are in the adult brain, testis, and ovaries. Mutant flies harboring a deletion of this gene have a dramatically reduced lifespan, and aberrant responses to odorants. RNA sequencing (RNA-seq) analysis identified several misregulated pathways, including immune genes and odorant-binding proteins. Importantly, lifespan could be restored in the mutant fly by neuronal expression of the mammalian MPPED2 ortholog. Since we could detect expression of the mammalian proteins in cultured neurons, our findings will aid in identifying the pleiotropic role(s) of the mammalian proteins.

## Results

### dMpped is a metallophosphodiesterase expressed during development and in multiple adult tissues

*CG16717* is located on chromosome 3 at cytogenetic location 67C4. The *CG16717* gene has two exons that are spliced to yield a single transcript, encoding for a protein of 300 amino acids. The entire protein coding sequence is present in exon 2 (http://flybase.org/reports/FBgn0036028.html). Since *CG16717* is the only ortholog of the MPPED1/MPPED2 family of proteins in *Drosophila*, we will henceforth refer to it as *dMpped* (*Drosophila* **m**etallo**p**ho-s**p**ho**e**sterase **d**omain containing). Sequence alignment of dMpped, hMPPED1 (human MPPED1), hMPPED2 (human MPPED2) and rMpped2 (rat MPPED2) reveals that dMpped shares ∼50% sequence similarity to the mammalian orthologs (Fig 1A). All the critical residues required to classify dMpped as a metallophosphoesterase are conserved including the metal binding residue aspartate at amino acid position 49 (D49) and histidine at position 51 (H51; H67 in MPPED2) [23] (Fig 1A).

We expressed and purified dMpped to determine its biochemical properties. The purified protein migrated as a protein of 37 kDa corresponding to the predicted molecular weight (Fig 1B). Purified dMpped was tested for phosphoesterase activity against a panel of colorigenic substrates, and like its mammalian orthologs rMPPED1 and rMPPED2 [9], dMpped, in the presence of $Mn^{2+}$ as the metal cofactor, showed phosphodiesterase activity and no detectable phosphomonoesterase activity (Fig 1C). dMpped hydrolyzed TmPP poorly, but not pNPPC, indicating some degree of specificity towards the kind of phosphodiester substrates it utilizes [9]. The enzyme could use several divalent cations, including $Ni^{2+}$ (S1A Fig). In contrast, MPPED2 showed little detectable activity with $Ni^{2+}$ [9]. The $K_m$ for $Mn^{2+}$ and pNPPP was 1.8 mM and ∼10 mM, respectively, comparable to that of MPPED2 [9] (S1B and S1C Fig). dMpped was active against cyclic nucleotides and hydrolyzed 2'3'cAMP more efficiently than either 3'5' cAMP or 3'5' cGMP (S1D Fig).

modEncode data indicated that *dMpped* was expressed at all stages of development and in various tissues (Fig 1D). To explore the expression profile of *dMpped* we performed RT-qPCR. We observed that dMpped is expressed through development from embryos to adult flies (Fig 1E) and in the testis and ovary (Fig 1F). We noted high expression in the brain (Fig 1F), reminiscent of the reported expression of mammalian *MPPED1* and *MPPED2* transcripts in the brain. Expression was similar during aging (S2A Fig).

We utilized genetic approaches to monitor the expression of the *dMpped* gene, using *IT-gal4^{1111-G4}* flies [26]. These flies have the *GAL4* coding sequence inserted within the only intron of the *dMpped* gene (Fig 1G). Therefore, transcription is expected to be under the

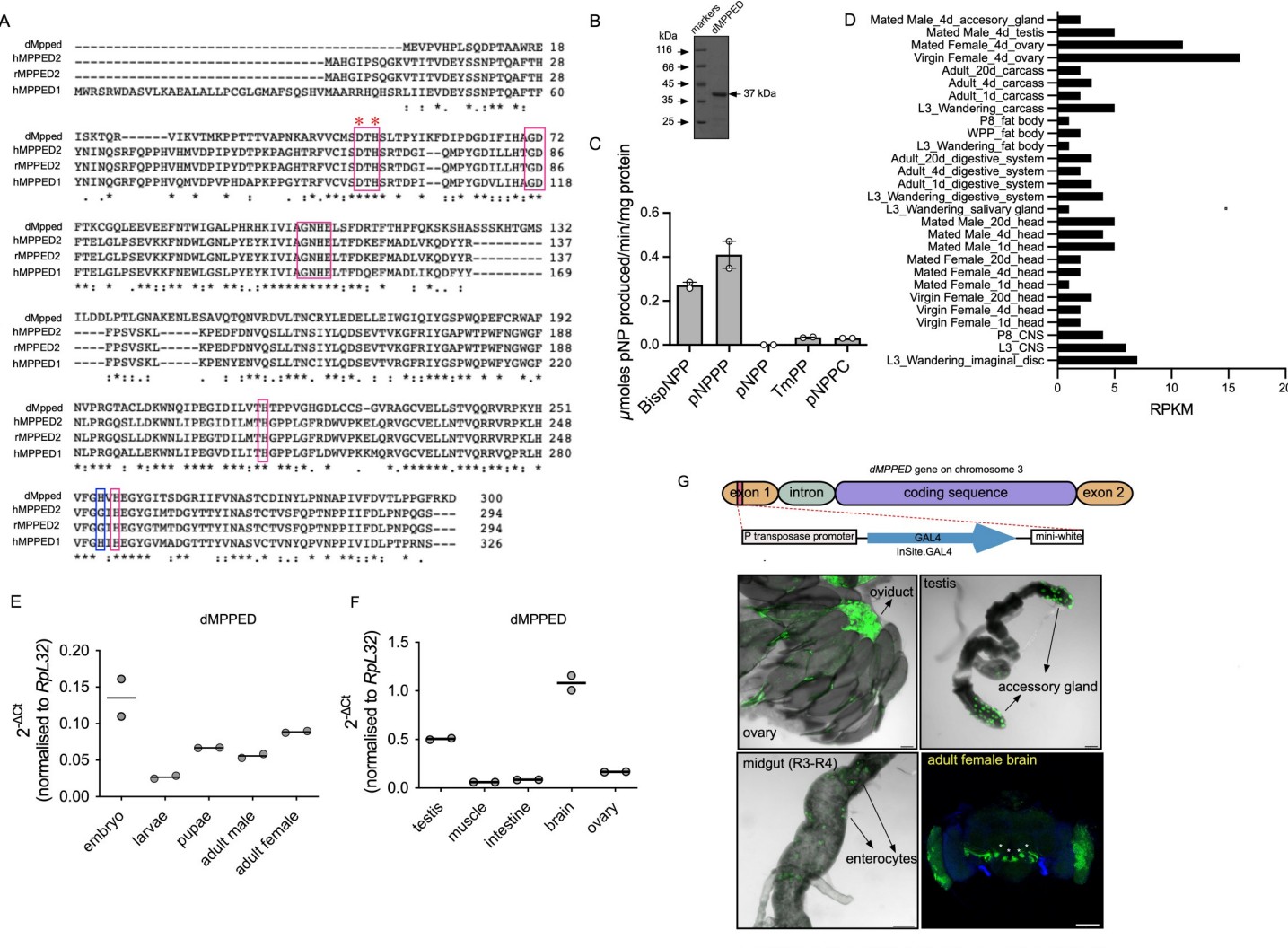

**Fig 1. dMpped is a metallophosphoesterase and its expression is enriched in neurons.** (A) Sequence alignment of dMpped with human MPPED1, human MPPED2 and rat MPPED2. Red boxes highlight the conserved residues characteristic of metallophosphoesterases. * indicates residues D49 and H52 mentioned in the text. Highlighted in the blue box is the histidine residue that distinguishes MPPED1 from MPPED2. (B) A Coomassie R stained gel showing purified dMpped protein used for biochemical assays. (C) The catalytic activity of dMpped with the indicated colorigenic substrates (10 mM), in the presence of $Mn^{2+}$ (5 mM) with dMpped (500 ng protein). Values represent the mean ± S.E. of duplicate determinations of experiments performed using two independent protein preparations. BispNPP, bis(p-nitrophenyl) phosphate; pNPPP, p-nitrophenyl phenylphosphonate; pNPP, p-nitrophenyl phosphate; TmPP, thymidine 5'-monophosphate-p-nitrophenyl ester; pNPPC, p-nitrophenylphosphoryl-choline. (D) Expression of *CG16717* (*dMpped*) obtained from an analysis of RNAseq data from the modEncode data hosted in Flybase ([www.flybase.org](www.flybase.org)). Transcript levels are seen at low levels in many tissues, including the CNS and larvae (E) Expression pattern of *dMpped* across developmental stages of the fly. Embryos (2 h after egg laying; 50), third instar larvae (wandering; 10), pupae (21 h after pupae formation; 10), male and female flies (3 days old; 10 each) were collected and RT-qPCR was performed. *dMpped* transcript levels have been normalized to *RpL32* transcript levels. The graph represents mean from 2 sets of samples collected independently. (F) Expression pattern of *dMpped* in different adult tissues. Testis and ovaries were collected from 50 male and female flies, respectively. The intestine (gut) was dissected from ~ 50 flies in total, and 100 flies were used for muscles and brains. The graph represents mean from 2 sets of samples collected independently. (G) Confocal images of the ovary, testis, posterior midgut (R3-R4) and brain in *pBAC(IT.GAL4)CG16717/UAS-mCD8GFP*. *pBac(IT.GAL4)* enhancer trap element is positioned within the *CG16717* gene. Ovaries show expression in the oviduct, the testis in secondary cells, and scattered enterocytes are GFP-positive in the posterior midgut. In the brain, strong expression is seen in the antennal lobe, marked with white asterix and in the optic lobe. The scale bar indicates 100 μm. 3D renderings of the brain sections of female brains are shown in S1 Movie, and enlarged images are shown in S1 Fig.

control of the same genetic elements and enhancers as *dMpped*. We crossed these flies to *UAS-mCD8-GFP* flies, adult tissues were dissected and imaged. In agreement with the RT-qPCR

data, GFP expression was detected in the adult ovaries in the oviduct, accessory glands of the testis, and low levels in enterocytes in the intestine (Fig 1G).

We have used female flies throughout this study in order to avoid differences in behavioral responses that we described later. Expression in female brains was localized to specific neurons in the brain (S2B Fig; [27]). These neurons could represent olfactory sensory neurons, neurons involved in thermosensation and hygrosensation [28] and/or projection neurons which project from the antennae to the antennal lobe, where they innervate specific glomeruli. Female flies demonstrated significant expression in the optic lobes (S2C Fig).

## Generation of of *dMpped^KO* flies and RNAseq analysis

To determine the functional role of *dMpped* in *Drosophila*, we generated a null allele (*dMpped^KO*) using ends-out homologous recombination [29–31] (S3A Fig). The gene deletion was verified by genomic PCR and RT-PCR (S3B and S3C Fig). In addition, we generated an antibody to dMpped, and western blotting confirmed the absence of protein in the heads of *dMpped^KO* flies (S3D Fig). To rule out off-site target effects on adjacent genes, we monitored the expression levels of *α-tubulin* and *furry* and observed that they were not affected in *dMpped^KO* flies (S3E Fig).

*dMpped^KO* flies were homozygous viable and showed no obvious developmental or phenotypic differences compared to wildtype flies. Since the role of dMpped in flies is unknown, we adopted an unbiased approach by identifying misregulated genes in *dMpped^KO* flies by RNAseq [32]. RNAseq analysis of 3-day old, virgin female flies revealed several differentially regulated genes in *dMpped^KO* flies (Fig 2A), with 592 up-regulated genes and 260 down-regulated genes at q-value < 0.01. Most of the genes were of unknown function, but it was interesting to note that several misregulated genes were co-expressed with dMpped in various tissues (Fig 2B).

A large set of genes important in defense response to bacteria (*Diptericin*, *Dpt; Drosomycin*, *Drs;* and *Drosocin* were upregulated in *dMpped^KO* flies (Fig 2A and 2C). Inflammation is a hallmark of faster aging and shorter lifespan [33–35]. Interestingly, *dMpped* is one of the major genes upregulated in S2 cells following infection with *Drosophila* C virus [36]. Thus, these findings appear to place dMpped as a link between innate immune pathways and life span in the fly.

Indeed, the life span of *dMpped^KO* flies was reduced when compared to *w^1118* flies (Fig 2D). Changes in levels of genes in the IMD and Toll pathways could account for the upregulation of antimicrobial peptides (AMPs). IMD associates with the Fas-associated death domain protein (FADD), that recruits the caspase-8 homolog death related ced-3/Nedd2-like caspase, DREDD. DREDD cleaves IMD, allowing interaction with Drosophila inhibitor of apoptosis-2 (dIAP-2), that ubiquitinates and stabilizes IMD. Recruitment of transforming growth factor β (TGF-β)-activating kinase 1 (TAK1) mediates phosphorylation of the IκB kinase (IKK) and Jun nuclear kinase (JNK). IK phosphorylates the N-terminal domain of Relish (Rel, the NFκB ortholog in flies), which, following DREDD-mediated cleavage of the C-terminus of Rel, allows nuclear migration of Rel and the induction of AMP transcription. To determine if the overexpression of AMPs contributed to life span reduction, we cross *dMpped^KO* with *Dredd* hypomorph flies. Life span was restored in *dMpped^KO* flies (Fig 2D), and a reduction in transcript levels of AMPs was seen (Fig 2E). Thus, overexpression of AMPs in the *dMpped^KO* flies could contribute to the reduced life span in flies, as suggested in earlier studies [37,38].

## Neuronal expression of dMpped regulates fly lifespan

Given that *dMpped* is expressed at high levels in the brain, and neuronal regulation of lifespan is well studied [37,39–41], we used a pan-neuronal driver, *elav-GAL4*, to express dMpped in *dMpped^KO* flies, after backcrossing flies used in experiments for 9 generations against *w^1118*

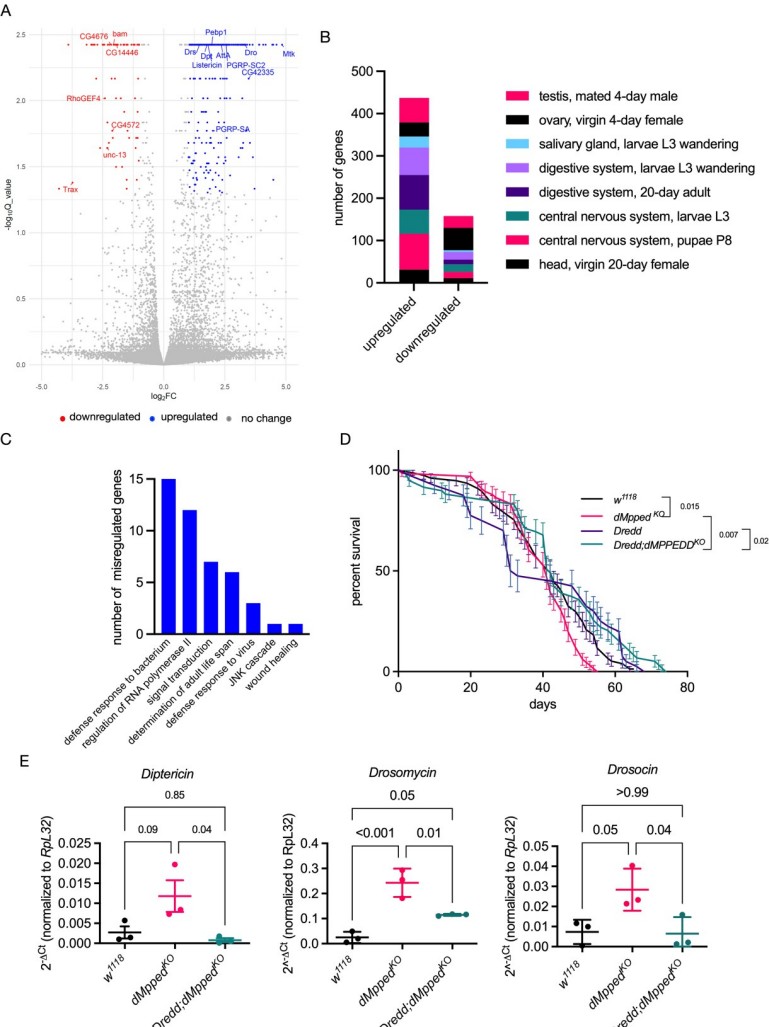

**Fig 2. RNA-seq analysis and misregulation of genes associated with defense pathways.** (A) A volcano plot showing protein-coding genes altered in 35-day old *dMpped*$^{KO}$ flies. The Log$_2$ fold change (Log$_2$FC) values are plotted on the x-axis and the negative log (base 10) of the Q value (-Log$_{10}$Q_value) on the y-axis. Blue (upregulated) and red (downregulated) dots represent genes with Q-values <0.05 and Log2 fold change values of more or less than one-fold. (B) The tissue distribution of differentially regulated genes. Misregulated genes (852, with 592 up-regulated and 260 down-regulated) were analysed by DGET (http://www.flyrnai.org/tools/dget/web/) based on published analysis of the Drosophila transcriptome [88]. (C) KEGG analysis of genes with Log2FC >2. Apart from several genes of unknown function, many misregulated genes were associated with defense response and immunity. (D) *dMpped*$^{KO}$ flies were crossed with *Dredd* hypomorph flies (*P[39]DreddEP1412 w*$^{1118}$) and life span monitored in virgin female flies. Flies (*w*$^{1118}$, n = 76; *dMpped*$^{KO}$ n = 66; *Dredd*, n = 40; *Dredd;dMpped*$^{KO}$, n = 59) were from at least three independent experiments. Log-rank test was used to compare survival across genotypes. Values shown at each time point are the mean ± SD, and the line represents the average across all experiments. p values are shown and compare the average life span across all three replicates of each genotype. (E) RT-qPCR of *AMPs* from RNA prepared from whole flies. Reducing *Dredd* activity in *dMpped*$^{KO}$ flies reduced expression the *AMPs*, indicating that *dMpped*$^{KO}$ flies showed misregulation of the IMD pathway. Each data point represents RNA prepared from 10 female flies, and values shown are the mean ± SD of three independent experiments. Data were analyzed by ANOVA and corrected for multiple comparisons using Tukey's test. p values are indicated across genotypes.

flies. This was done to ensure no other mutations were present in the *dMpped*$^{KO}$ flies in subsequent experiments that describe phenotypes related to life span with comparisons made to *w*$^{1118}$ flies. We confirmed expression of dMpped by western blotting of protein extracts

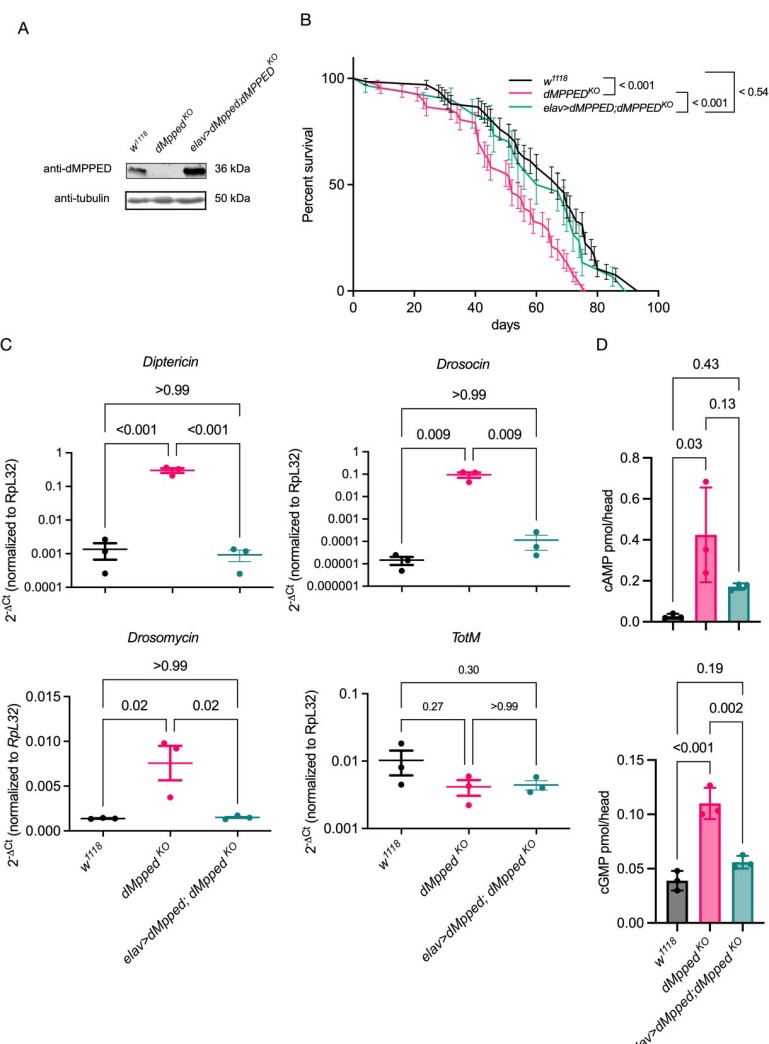

**Fig 3. Neuronal expression of dMpped determines fly lifespan, AMP expression and cyclic nucleotide levels in the brain.** (A) Western blot to confirm that *dMpped*[KO] is a protein null allele, and expression of dMpped in the brain of flies where dMpped expression is driven in the neurons. The blot was re-probed with an anti-tubulin antibody to normalize protein loading. (B) Survival curves of backcrossed flies of indicated genotypes. Flies (*w*[1118], n = 66; *dMpped*[KO], n = 66; *elav-Gal4>dMpped; dMpped*[KO], n = 30) were from at least three independent experiments. Log-rank test was used to compare across genotypes. Values shown at each time point are the mean ± SD, and the line represents the average across all experiments. p values are shown and compare the average life span across all three replicates of each genotype. (C) RT-qPCR to confirm upregulation of *AMP*s in *dMpped*[KO] flies. Levels were restored following neuronal expression of dMpped in *dMpped*[KO] flies. Each data point represents RNA prepared from 10 female flies, and histograms correspond to the mean value ± SD of three independent experiments. Data were analyzed by ANOVA and corrected for multiple comparisons using Tukey's test. p values are indicated across genotypes. (D) Cyclic AMP and Cyclic GMP levels were measured in the heads of flies of the indicated genotypes. Levels of cAMP and cGMP were significantly higher in *dMpped*[KO] flies, correlated with reduced phosphodiesterase activity in flies deleted for dMpped. Extracts were prepared from 10 female fly heads, and histograms correspond to the mean value ± SD of three independent experiments. Data were analyzed by ANOVA and corrected for multiple comparisons using Tukey's test. p values are indicated across genotypes.

prepared from the brain (Figs 3A and S4). Further, restoration of dMpped expression in neurons was sufficient to rescue the reduced lifespan of *dMpped*[KO] flies (Fig 4B).

We then measured levels of AMPs in the head of wildtype, *dMpped*[KO] and neuronally rescued flies and found that the elevated levels of *Diptericin*, *Drosocin* and *Drosomycin* were

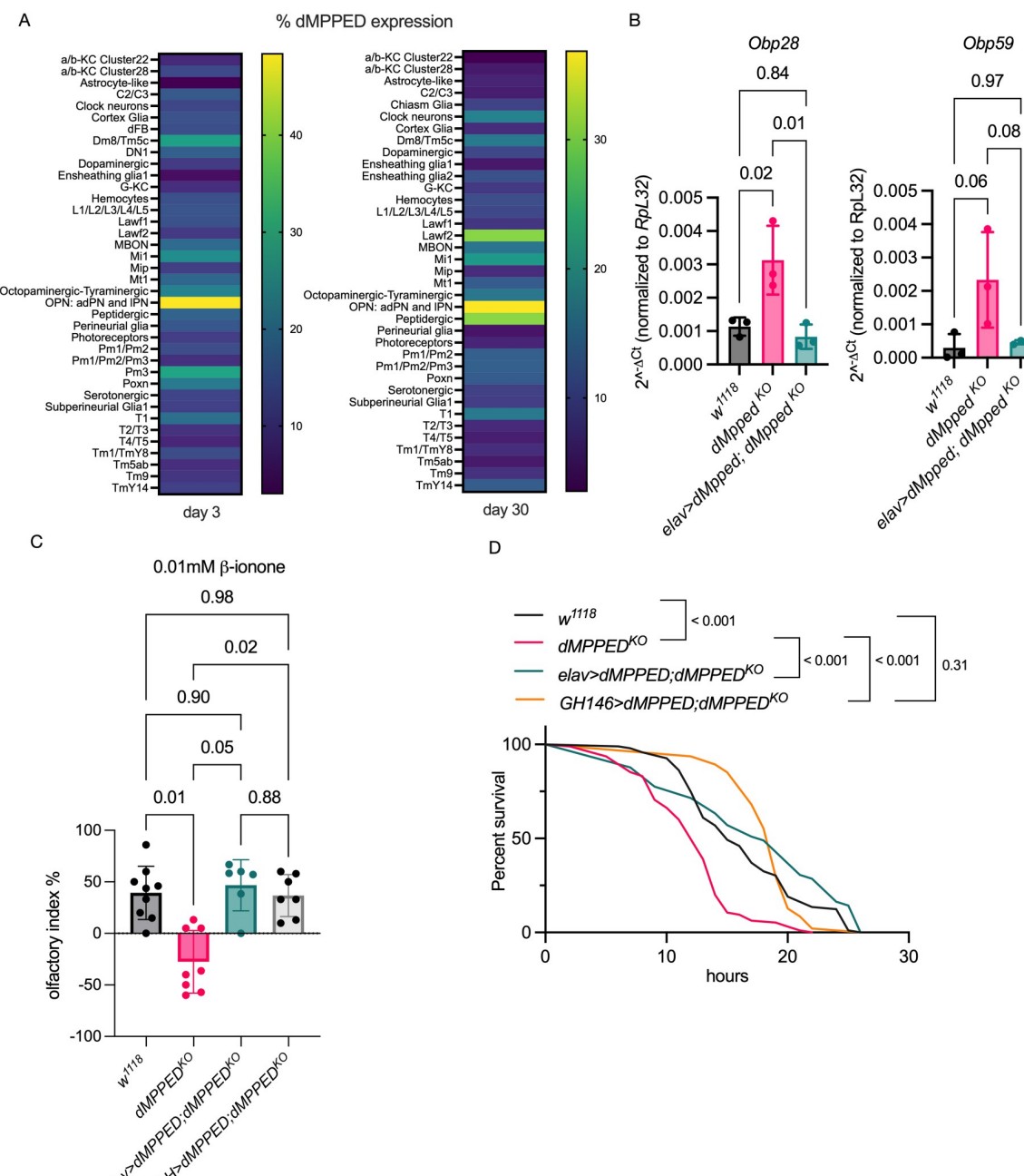

**Fig 4. *dMpped* regulates expression of Obps and affects odorant-driven behavior and tolerance to desiccation.** (A) Expression of dMpped in specific neurons in the fly brain. Data was extracted from single cell RNAseq analysis [47] and shown as a heat map indicating expression levels in flies of 3 and 30 days of age. The highest expression is seen in OPNs in agreement with Fig 1G. (B) RT-qPCR of *Obp28* and *Obp59* indicating misregulation in *dMpped^KO* indicating misregulation in *dMpped^KO* flies with levels restored to that seen in control flies following neuronal expression of dMpped in *dMpped^KO* flies. At least 10 flies were used for each experiment and data shown is across 3 independent experiments. Data were analyzed by ANOVA and corrected for multiple comparisons using Tukey's test. Values are mean ± SD and p values across genotypes are shown. (C) Response of flies to varying concentrations of β-ionone, an attractant that binds to Obp28. *dMpped^KO* flies show no preference in the Y-maze test at low concentrations (0.01mM) of β-ionone, and the behavioral response was restored in *dMpped^KO* flies with the expression of *dMpped* restored pan neuronally or specifically in olfactory projection neurons. Flies (*w^1118*, n = 104; *dMpped^KO*, n = 89; *elav-Gal4>dMpped;* dMpped^KO*, n = 44; *GH>dMPPED;dMPPED^KO*, n = 64) were from at least six independent experiments. Data were analyzed by ANOVA and corrected for multiple comparisons using Tukey's test. Values are mean ± SD and p values are shown. (D) Flies were subjected to desiccation and the time taken for to die was monitored. Restoration of expression of dMpped in *dMpped^KO* flies restored desiccation resistance to that seen in wildtype flies. Flies per genotype are pooled from at least three independent experiments. Log-rank test used for comparing wild type (*w1118*, n = 92); *dMpped^KO* (n = 95) flies; *elav-Gal4>dMpped; dMpped^KO* (n = 49) and *GH146>dMPPED;dMPPED^KO*, n = 50. p values across genotypes are shown in the graph.

restored to that seen in wildtype flies (Fig 3C) and were correlated with an increased life-span in *elav>dMpped;dMpped*[KO] flies (Fig 3B). Levels of Turandot M (*TotM*) transcript, which is a target of the Jak/Stat pathway and also induced in response to bacterial infection [42], were unchanged in *dMpped*[KO] flies, indicating that dMpped specifically modulated the IMD and Toll pathways.

Metallophosphoesterases are promiscuous in their substrate utilization, but the diesterases specifically cleave only molecules with diester bonds. Since dMpped could hydrolyze cAMP and cGMP (S1D Fig), we measured both cAMP and cGMP levels in the brains of wildtype, *dMpped*[KO], and *dMpped*[KO] flies expressing dMpped in neurons. We detected higher levels of both cyclic nucleotides in the brains of *dMpped*[KO] flies (Fig 3D), which was restored on neuronal expression of dMpped. Interestingly, cyclic nucleotide levels are associated with life span in Drosophila. *Dunce* encodes a cAMP phosphodiesterase, and knock-out female flies (which should harbor elevated cAMP levels) show a reduction in life span [43], in agreement with our results. Cyclic nucleotides can bring about their action by activating cAMP or cGMP-dependent kinases. Interestingly, reduced levels of the cGMP-dependent protein kinase (PKG) in flies increase life span [44]. Further, suppression of neuronal cGMP levels in *C. elegans* results in extended life-span in a FOXO-dependent manner [45]. Therefore, we propose that neuronally elevated cGMP in *dMpped*[KO] flies may also contribute to life span regulation in the fly.

## Attractant perception is compromised in *dMpped*[KO] flies

Among the genes significantly upregulated in *dMpped*[KO] flies in the RNAseq analysis were olfactory binding proteins (Obps; S5 Fig), a diverse group of proteins that show odorant binding *in vitro* but also have a broader role in insects than previously envisaged [46]. There are 52 Obps localized mainly to the sensilla found in insect antennae and are thought to transport hydrophobic odorants across the aqueous sensillar lymph to olfactory receptors. We noted that from single cell RNAseq analysis of the Drosophila brain [47], *dMpped* was expressed at high levels in the olfactory anterodorsal, lateral, and ventral lineage projection neurons (Fig 4A) in agreement with our expression data in the antennal lobe (Fig 1F). Furthermore, an increase in levels of *dMPPED* transcripts are seen in OPNs on day 30 and in peptidergic neurons (Fig 4A). Peptidergic neurons control a number of processes including metabolism, circadian timing, and cues that regulate food search, aggression and mating [48]. Therefore, increased expression of dMPPED in these neurons could regulate these functions in aging flies.

We focused on the implications of the overexpression of two Obps in *dMpped*[KO] flies, Obp28 and Obp59. Obp28a binds the floral odor attractant β-ionone with micromolar affinity, and deletion of the Obp28a gene resulted in reduced olfactory preference at low concentrations of β-ionone [49]. We first validated the overexpression of Obp28 in *dMpped*[KO] flies by RT-qPCR (Fig 4B) and saw that expression of dMpped in the neurons of *dMpped*[KO] flies could restore transcript levels of *Obp28* to those seen in control flies. We then tested the ability of flies to move towards β-ionone in a Y-shaped olfactometer [49] at two different concentrations of β-ionone. While attraction to the odor was similar in control and *dMpped*[KO] flies at the higher (0.05 μM) concentration (S5B Fig), we observed that at low concentrations of β-ionone (0.01 μM) *dMpped*[KO] flies showed no preference for a movement toward the odorant-containing arm. These results paradoxically mimic those seen in Obp28a-deleted flies, suggesting that at low concentrations of β-ionone, Obp28a-mediated responses depend on a balanced expression of Obp28. Alternatively, one could speculate that the interaction of Obp28 with the receptor could depend on the presence of neuronally expressed dMpped, either by direct interaction with the Obp or its receptor, and this interaction could be altered in the absence of dMPPED.

To confirm that the behavior towards odorants in $dMpped^{KO}$ flies was explicitly in response to β-ionone and mediated by a unique set of neurons, we tested the behavior of $dMpped^{KO}$ flies to ammonia and acetophenone. High ammonia concentrations are repellant to wild-type flies (S5C Fig). Ammonia at a lower concentration is a strong attractant for flies, and attraction behavior is mediated by olfactory sensory neurons that express ionotropic receptor IR92a [50] and do not require an Obp. Acetophenone is a repellant and induces behavioral responses mediated via *Obp56f*, *Obp56h*, and *Obp83a* [51], none of which were misregulated in $dMpped^{KO}$ flies (S5A Fig). We observed that $dMpped^{KO}$ flies responded similarly to control flies with these odorants at concentrations tested in earlier studies [50,51] (S5C and S5D Fig).

We then focused on Obp59a, upregulated $\sim 4.5$-fold in the RNAseq analysis (S5A Fig). Loss of Obp59a leads to an increase in desiccation resistance [52]. We validated the upregulation of *Obp59* by RT-qPCR in $dMpped^{KO}$ flies, and expression was restored to control levels on expression of dMpped in the neurons of $dMpped^{KO}$ flies (Fig 4B). In agreement with the results seen in Obp59 knockout flies, we saw increased sensitivity to desiccation in $dMpped^{KO}$ flies, which was complemented by reducing levels of Obp59a following neuronal expression of dMpped (Fig 4D).

To rescue the expression of dMPPED in olfactory neurons, we used the *GH146-Gal4* driver line, which labels a broad set of second-order olfactory projection neurons with little background expression [53]. We tested olfaction at low concentrations of β-ionone (0.01 μM) in $dMpped^{KO}$ flies expressing *dMPPED* in olfactory neurons using the *GH146-Gal4* line. We found that the expression of *dMPPED* in olfactory projection neurons allowed them to sense 0.01 μM β-ionone (Fig 4C). Further, these complemented flies were not as sensitive to desiccation as the $dMpped^{KO}$ flies (Fig 4D). Therefore, expression of dMPPED in olfactory neurons is required to allow specific odorant perception and optimum response to desiccation stress.

In conclusion, the expression of dMpped in possibly very specific neurons alters odorant perception and hygrosensation in flies, emphasizing the pleiotropic roles of dMpped in the physiology of the fly.

## The mammalian ortholog of dMpped restores life span in $dMpped^{KO}$ flies

The MPPEDs are highly evolutionarily conserved proteins across metazoans and the sequence similarity between the fly and the mammalian MPPED proteins is $\sim 50\%$ (Fig 1A). The reduced lifespan of $dMpped^{KO}$ flies offered an excellent model to test possible functional conservation between dMpped and its mammalian ortholog, MPPED2. We first monitored the expression of *Mpped1* and *Mpped2* in mouse tissues by PCR and observed the expression of both these genes in the ovary, kidney, and various brain regions (Fig 5A). Interestingly, *Mpped2* was also expressed in the heart. To confirm protein expression in the brain, we performed western blot analysis with a monoclonal antibody raised to Mpped2 and showed that this antibody could recognize both Mpped1 and Mpped2 (Fig 5B). We could detect robust protein expression (s) in brain homogenates and higher levels in the cortex compared to the hippocampus (Figs 5B and S5). Two bands of molecular weight 33 kDa and 25 kDa could be observed. A perusal of the Allen Brain Atlas (http://mouse.brain-map.org) indicated that expression of Mpped1 was highest in the isocortex, olfactory areas, hippocampal formation, and cortical subplate. No data was provided for Mpped2. However, based on the molecular weight predicted of Mpped1 and Mpped2, either Mpped2 is the major protein expressed in the brain, or post-translational proteolysis of Mpped1, initiation of translation of rodent Mpped1 at Met25 or Met31 (Fig 1A), or post-translational proteolysis of both proteins cannot be ruled out.

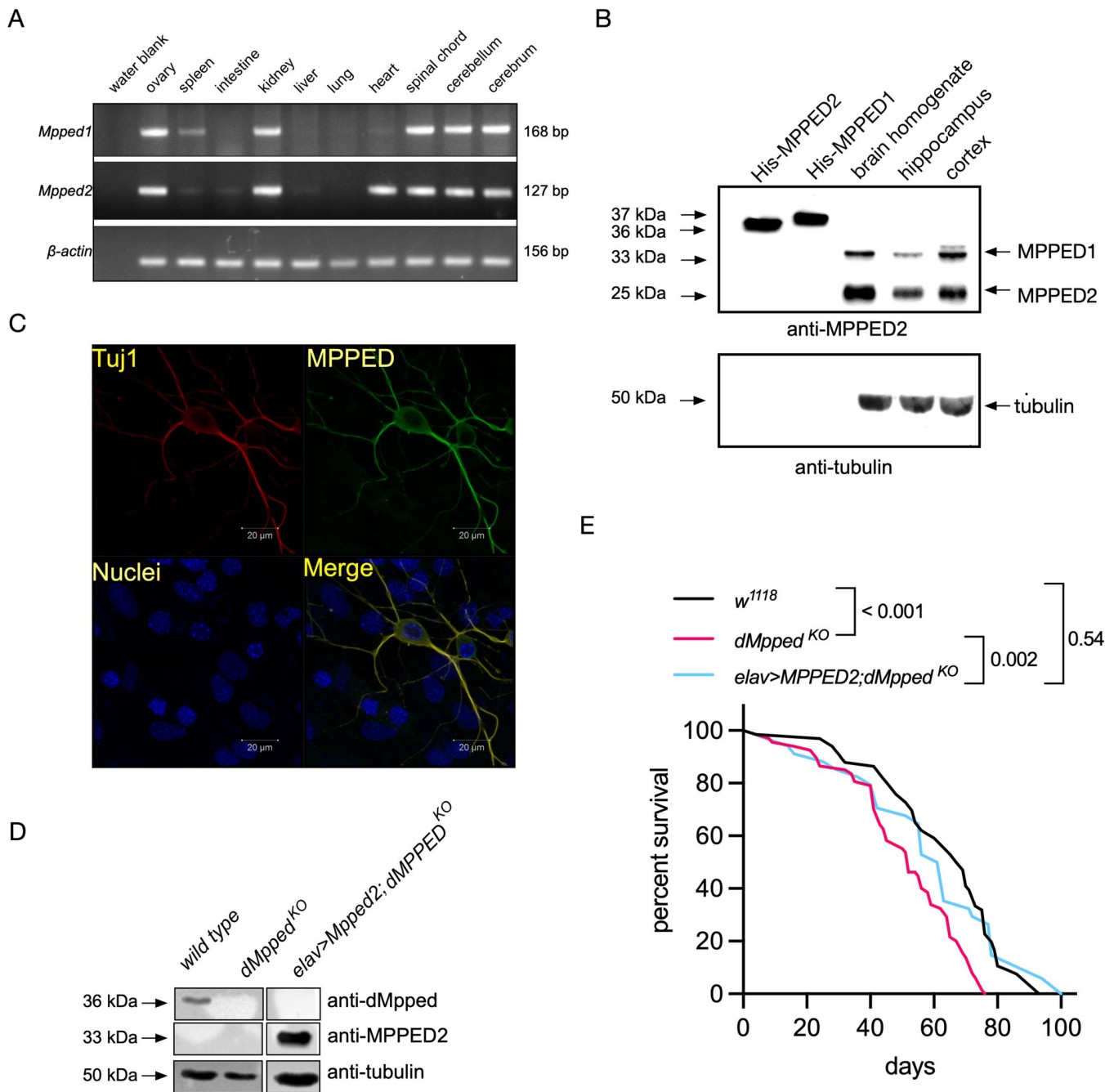

**Fig 5. Mammalian ortholog of dMpped regulates lifespan in flies.** (A) Expression of mMPPED1 and mMPPED2 in mouse tissues. RNA was prepared from the indicated tissues, and expression of mMPPED1, and mMPPED2 monitored following reverse transcription and PCR using specific primers. Expression of β-actin was used to check equivalent cDNA synthesis across tissues. (B) Western blot analysis with purified mMPPED1, mMPPED2 and lysates prepared from total mouse brain or homogenates prepared from the hippocampus and cortex using a monoclonal antibody raised to rat MPPED2. Blots were probed with the MPPED monoclonal antibody and protein loaded in each lane normalized to tubulin. Recombinant, purified, histidine-tagged mMPPED1 and mMPPED2 were used as controls and migrate higher than the endogenous protein present in brain lysates due the hexahistidine tag at the N-terminus of the proteins that facilitated purification. Data is representative of experiments performed with two independently prepared homogenates. The two bands seen in brain homogenates migrate at sizes predicted for mMPPED1 and mMPPED2. (C) Immunocytochemical analysis to show expression of mMPPED in neurons prepared from the cortex of 1-day old mice. (D) Western blot performed homogenates prepared from the brain of flies of the indicated genotypes. Blots were probed either with the monoclonal antibody raised to mammalian Mpped2 or the polyclonal antibody raised to dMpped. Protein loading was normalized by

using an antibody to tubulin. The data shown is representative of experiments performed with independently prepared homogenates at least twice. (E) Flies ($w^{1118}$, n = 66; dMppedKO, n = 66; *elav-Gal4>MPPED2; dMpped$^{KO}$*, n = 34) were from three independent experiments. Log-rank test was used for comparing across genotypes. p-values across genotypes are shown in the Figure.

We prepared neuronal cultures from the mouse cortex, and immunofluorescence indicated co-expression of Tuj1(expressed only in the neurons) and Mpped1 and/or Mpped2 (Fig 5C). This is the first demonstration of the expression of these proteins in mammalian neurons.

To determine whether mammalian Mpped2 could restore life span in flies, we expressed rat MPPED2 neuronally in *dMpped$^{KO}$* flies and confirmed expression in the fly brain by western blot analysis (Fig 5D). Interestingly, lifespan in flies expressing MPPED2 was restored to wild-type levels in *dMpped$^{KO}$* flies (Fig 5E). Therefore, mammalian MPPED2 has functional conservation with dMpped, suggesting that the fly could be used to delineate further the mechanisms by which MPPED2 and/or MPPED1 regulate diverse pathways in different mammalian tissues.

## Discussion

Here, we present the first characterization of the fly ortholog of mammalian MPPED1/ MPPED2 proteins and show how its neuronal expression regulates lifespan, odorant perception and hygrosensation in the adult fly. While several genes that regulate lifespan in the fly are also critical for normal development, dMpped is not essential during development but plays a role in aging in the adult fly. dMpped can hydrolyze 2'3'-cAMP to 3'AMP like the mammalian orthologs (S1D Fig). CNPase hydrolyzes 2'3'-cAMP in mammals, and accumulation of this cyclic nucleotide leads to an increased susceptibility to brain injury and neurological disease [54]. Whether an additional role of dMpped relates to its catalytic activity and whether 2'3' cAMP plays a role in the fly brain remains to be investigated.

The importance of cAMP in dopamine-mediated signaling in the mushroom body [55], and odor-induced cAMP production in olfactory sensory neurons in the antenna plays a central role of olfactory conditioning in Drosophila [56]. Single-cell RNAseq data showed low expression of *dMpped* in dopaminergic neurons (Fig 4A) but robust expression in olfactory projection neurons. Projection neurons respond to a broader range of odors than their corresponding olfactory receptor neurons. It is conceivable that the elevated levels of cAMP/cGMP in these neurons in *dMpped$^{KO}$* flies could modify output and memory in an odor-specific manner.

There is increasing evidence that the aging immune system can lead to low-grade inflammation, which is associated with increased mortality. We have shown an unexpected role of a metallophosphoesterase in controlling lifespan by modulating the levels of AMPs during aging. The higher levels of AMPs detected in whole *dMpped$^{KO}$* flies could originate from the fat body following innervation by neurons in which dMpped is expressed. In the Malpighian tubules, increased cGMP levels can lead to the translocation of the transcription factor Relish to the nucleus and activation of the IMD pathway [57]. Therefore, elevated levels of cAMP and cGMP in the heads of *dMpped$^{KO}$* flies could lead to activation of IMD in a cell non-autonomous manner in the fat body fragments present near the brain.

Is there a link between odorant-binding proteins and AMP production? Growing evidence suggests that Obps are expressed in several tissues, including hemocytes from the mosquito and fat bodies [58]. We have observed that specific Obps are expressed in hemocytes prepared from Drosophila after sterile injury [59]. A recent study has shown that an *Obp28a* mutant Drosophila line perished faster following sterile thoracic injury, with lower melanin deposition at the wound site [60]. Further, expression of *Obp28a* was significantly reduced in larvae reared axenically, while conventionally reared flies with endogenous microbiota showed increased expression of *lozenge* (a gene that controls the lineage specification of prohemocytes

into crystal cells that release melanin at the site of injury)[61]. Crystal cells are important in Drosophila innate immunity and stress responses, even to gaseous chemicals such as $CO_2$ [62]. Therefore, the production of AMPs may be linked to *Obp* expression in *dMpped*[KO] flies, which in turn impacts on the reduced lifespan seen in these flies.

An additional role of odor perception and immunity lies in the finding that upon olfactory stimulation, specific olfactory neurons induce the secretion of GABA from a subset of neurosecretory cells. GABA then binds to metabotropic GABA$_B$ receptors expressed on blood progenitors signal causing high cytosolic $Ca^{2+}$, required for progenitor maintenance [63]. Since dMpped is expressed in olfactory neurons, the activation of these neurons could be altered in its absence, which in turn may alter hematopoiesis.

Expression of *Obps* is affected both by age and nutrient availability in flies [64]. An *Obp83b* knock-out strain is long-lived, and longevity in these flies appears to be largely controlled by a diet-independent pathway. Further, female flies were more resistant to hyperoxia and resistant to starvation. In *dMPPED*[KO] flies, *Obp83b* shows a trend of upregulation ($\sim$ 4-fold, p value 0.05), and this increase in expression could reduce the lifespan, based on studies with *Obp83* knock-out flies. Links between odorant perception and longevity need to be explored further, and the evolutionary conservation of the MPPED family suggests that such associations may also emerge in mammals.

Given the moonlighting activities of the metallophosphoesterase family of proteins [12,65], dMpped could serve as a scaffolding protein. From the analysis of the *Drosophila* interactome described by high throughput approaches [66], only one gene product, Sh3px1, was reported to interact with dMpped with some confidence. This protein is the single fly ortholog of the human sorting nexin 9 family known to function in vesicular sorting [67,68]. SH3PX1 has been found to regulate the formation of lamellipodia, tubules, and long protrusions in S2 cells [69]. In the fly, SH3PX1 localizes to neuromuscular junctions where it regulates synaptic ultrastructure [70]. Neurotransmitter release was significantly diminished in SH3PX1 mutants and functional interactions with Nwk, a conserved F-BAR protein that attenuates synaptic growth and promotes synaptic function in *Drosophila*, were observed [71]. Interaction with Nwk was via the SH3 domain of SH3PX1, which recognizes proline-rich sequences. The co-expression of dMpped and SH3PX1 in neuronal tissue may allow for functional interaction that could modulate Nwk action, since the SH3 domain in SH3PX1 could interact with the PXXP motif in dMpped (residues 4–7; Fig 1A) and sequester it from Nwk.

Recently, a role for Sh3px1 in regulating the innate immune response was described that was mediated by its interaction with the autophagy protein, Atg8a and Tyk/Tab2 [72]. Tak1 and Tab2, a co-activator of Tak1, interact with Atg8 and are selectively targeted for autophagy to regulate the IMD pathway. The predicted interaction of dMpped and Sh3px1 could indicate links between the regulation of the IMD pathway, autophagy machinery and dMpped.

dMpped shows a marginally higher sequence identity to hMPPED1 than hMPPED2 (48.7% to 47.6% respectively). Are there proteins that have been shown to interact with MPPED1/MPPED2 which may have implications as interacting partners of dMpped? In an extensive screen across the human genome involving affinity purification and mass spectrometry of interacting proteins (https://bioplex.hms.harvard.edu), the only common proteins that interacted with both MPPED1 and MPPED2 were transcription factors NR2F1 and NR2F6. These nuclear receptor factors are not conserved in *Drosophila*, which has far fewer nuclear-receptor genes than any other model organism [73]. However, a close homolog of these proteins in mammals is NR2F3, or COUP-TF1, which has the ortholog seven-up (*svp*) in the fly [73]. *svp* has crucial roles in neuronal development during embryogenesis and in the development of photoreceptor cells [74]. *COUP-TF1* is also required for neuronal development and axon

**Fig 6. Schematic showing pleiotropic roles of dMPPED.** Class III metallophosphoesterases are evolutionarily conserved and members of the MPPED family are neuronally expressed in the brain of mammals and flies. *dMPPED^KO* flies show misregulation of *Obps*, elevated levels of cAMP and cGMP and higher expression of *AMP*s. All these changes can influence longevity, with *dMPPED^KO* flies showing a decreased life span. The Figure was created with BioRender. com.

guidance [75]. Interactions, either direct or genetic, between dMpped and *svp* would be an interesting line of study in the future.

In summary, we have identified a new player in regulating several physiological responses, which may impinge on longevity in flies (Fig 6). The presence of orthologs of dMpped in higher animals suggests that the role of this protein in mammals is worthy of study. Our analysis reveals that this protein could serve as a focal point for interaction and cross-talk with several pathways, either through direct interaction or by modulating the activity of a few proteins, which could then impact more globally.

## Materials and methods

### Cloning and mutagenesis of dMpped

The full-length coding region of *dMpped* was amplified from cDNA prepared from whole flies using dMpped_MfeI_Fwd and dMpped_XhoI_Rvs primers (S1 Table), digested with XhoI

and cloned into EcoRV and XhoI digested pBKSII vector to generate pBKS-dMpped. The clone was verified by sequencing (Macrogen, South Korea). The MfeI-XhoI fragment from pBKS-dMpped was cloned into EcoRI and XhoI digested pPROExHT-B vector to obtain pPRO-dMpped. The same fragment was cloned into EcoRI and XhoI digested pUAST-attB vector to obtain pUAST-attB-dMpped.

## Expression and purification of dMpped

The pPRO clones of dMpped were transformed into *E. coli* BL21DE3 and proteins were expressed as described earlier [9]. Gel filtration of purified protein was carried out in buffer containing 50 mM Tris/HCl, 5 mM 2-mercaptoethanol, 50 mM NaCl, and 10% glycerol at pH 8.8 and 4˚C at a flow rate of 200 ul/min using a Superose 12 column and an AKTA fast protein liquid chromatography system (GE Healthcare). The protein eluates were stored in aliquots at -70˚C until further use.

## Biochemical assays

Enzyme assays for various activities (phosphatase, phosphodiesterase, nuclease and phospholipase) were performed in a triple buffer system (MES, HEPES, diethanolamine, 50 mM (pH 9.0)), 5 mM 2-mercaptoethanol, and 10 mM NaCl in the presence of 10 mM concentrations of the specified substrate and 5mM $Mn^{2+}$ as the metal cofactor. Assays were stopped by adding 10 ml of 200 mM NaOH, and absorbance was monitored at 405 nm. The amount of p-nitro-phenol formed was estimated based on its molar extinction coefficient of 18,450 $M^{-1}$ $cm^{-1}$.

Hydrolysis of cyclic nucleotides was measured using the malachite green assay [76]. Purified dMpped (5 μg) was incubated with either 2'3' cAMP (1mM), 3'5' cAMP (5 mM) or 3'5' cGMP (1 mM) in 50 mM of the triple buffer system (MES, HEPES and diethanolamine) [77] at pH 9.0, containing 10 mM NaCl, 5 mM β-mercaptoethanol, 5 mM $MnCl_2$, and 0.1U calf intestinal alkaline phosphatase in a final volume of 50 μl. The reaction was carried out at 37˚C for 15 min and terminated by the addition of 100 μl malachite green solution. Absorption at 620 nm was recorded and the amount of inorganic phosphate released (following the action of alkaline phosphatase on the products 5'/2'/3' AMP/GMP formed during the reaction) was interpolated from a standard curve, generated by using known amounts of inorganic phosphate.

## Fly culture

Unless mentioned otherwise, flies (*Drosophila melanogaster*) were reared reared using standard fly medium comprising of 8% cornmeal, 4% sucrose, 2% dextrose, 1.5% yeast extract, 0.8% agar, supplemented with 0.4% propionic acid, 0.06% orthophosphoric acid and 0.07% benzoic acid. Cultures were maintained at 25˚C and 50% relative humidity under 12h light/12h dark cycles. The wild type strain used was *w^1118*. All flies used in this study are listed in S2 Table.

Virgin female flies (0–3 day old) were collected and kept in groups of 10 flies per vial. The number of dead flies was recorded every 3 days, when flies were transferred to fresh media vials, for lifespan analyses. All experiments described here were performed at 25˚C.

## Dissection and imaging

Adult tissues were dissected in cold PBS and fixed in 4% paraformaldehyde (PFA) for 20 minutes at room temperature. Samples were then washed thrice with PBS containing 0.1% Triton X-100 for 5 min each and stained with Hoechst nucleic acid stain for 20 minutes. Samples

were mounted on a glass slide using Antifade solution and coverslips were sealed using nail-polish. Images were acquired using a Leica TCS SP8 confocal microscope.

## Generation of *dMpped*[KO] flies

A loss-of-function mutant was generated using ends-out homologous recombination [29]. For this, 4.5kb genomic region immediately upstream of the *dMpped* coding sequence and 4.5kb genomic region immediately downstream were amplified using fly genomic DNA as template and ExTaq polymerase (Takara). The two amplicons were sequentially cloned into the 5' and 3' multiple cloning sites of the pGX-attP vector [78], respectively, thus obtaining pGX-attP-dMpped. This construct was microinjected into *w*[1118] embryos to obtain P 'donor' flies [29]. These flies were used in a series of crosses and the progeny screened for loss of *dMpped* as described previously [78]. The *dMpped* knock-out thus obtained was further verified by genomic and RT-PCR.

The dMpped mutant, the driver elav-gal4 and UAS-dMpped were backcrossed nine times into w1118 to homogenize the genetic background.

### Lifespan

Virgin female flies (0–3 day old) were collected and kept in groups of 10 flies per vial. Flies were maintained at 25°C in an incubator set on a 12 h light/12 h dark cycle. The number of dead flies was recorded every 3 days, when flies were transferred to fresh media vials.

### Genomic and quantitative real-time PCR

For genomic DNA isolation, 10 flies were homogenized in 100 μL of buffer A (100 mM Tris-HCl (pH 7.5), 100 mM EDTA, 100 mM NaCl, 0.5% SDS). An additional 100 μL of buffer A was added and the samples were incubated at 65 $^{\circ}$C for 30 min. 400 μL of buffer B (1 part 5 M potassium acetate and 2.5 parts of 6 M lithium chloride) was added, and the mix was incubated on ice for 10 min. Following this, the debris was removed by centrifugation and the genomic DNA in the supernatant was precipitated using 300 μL of 2-propanol, washed with 70% ethanol, air dried, and resuspended in 20 μL of TE (10 mM Tris-HCl (pH 7.5), 1 mM EDTA) and genomic DNA was quantified by measuring the absorbance at 260 nm on a NanoDrop spectrophotometer (Thermo Scientific). Genomic PCR was performed using 50 ng of genomic DNA, 2.5 pmoles of gene-specific forward and reverse primers, 0.2 mM dNTPs, and 1 U of Taq DNA polymerase in a 20 μL reaction containing 1X standard Taq buffer.

RNA was isolated from flies of indicated age using the TRI reagent (Sigma). Real time quantitative PCR (RT-qPCR) was performed using the VeriQuest SYBR Green qPCR master mix with ROX (Affymetrix) on an ABI 7000 real time PCR machine (Applied Biosystems). Transcript levels of all the genes tested were normalized to transcript levels of ribosomal protein-49 (RpL32) using the ΔCt method wherein Ct stands for Cycle Threshold. Transcript levels have been plotted as $2^{-\Delta Ct}$ wherein $\Delta Ct = Ct_{gene} - Ct_{RpL32}$.

### Western blot

Purified dMpped protein was injected into rabbits to raise polyclonal antibodies to the protein. Polyclonal antibody against MPPED2 was available in the laboratory [9]. Fly brains were homogenized in 20 μL of homogenization buffer (50 mM Tris-Cl (pH 7.5), 2 mM EDTA, 1 mM DTT, 100 mM NaCl, and 1x Roche protease inhibitor mix). Samples were centrifuged at 13,000 g for 10 min, Laemmli sample buffer was added to the supernatant and boiled for 5 min. Protein samples were resolved on a 12% SDS-polyacrylamide gel and transferred onto a

PVDF membrane (Immobilion P, Millipore). The PVDF membrane was rinsed with TBST (10 mM Tris- HCl (pH 7.5), 100 mM NaCl, 0.1% Tween 20) and blocked for 1 h using 5% BSA made in TBST. Polyclonal anti-dMpped IgG (1 μg/mL) or anti-MPPED2 (culture supernatant from hybridoma at 1: 500 dilution) was added into the blocking solution, and the blot was incubated overnight at 4˚C. The membrane was washed and then incubated with TBST containing 0.2% BSA and horse radish peroxidase-conjugated anti-rabbit secondary antibody for 1h at room temperature. Bound antibody was detected by Immobilon Western chemiluminescent HRP substrate (Millipore) on the FluorChem Q MultiImage III system (Alpha Innotech). Anti-tubulin antibody (12G10; Developmental Studies Hybridoma Bank) (1:1000) was used to detect tubulin, serving as the loading control for individual lanes.

## RNA-Sequencing

10 virgin wild type and $dMpped^{KO}$ female flies (35 days old) were anaesthetized and collected. The collection and RNA extraction was done in triplicates. RNA extraction was performed with the help of RNeasy kit (Qiagen) as per manufacturer's protocol, DNased, and quantified using Nanodrop. 1.5 μg of RNA was subjected to RNA-Sequencing (Genotypic Technology Pvt. Ltd, Bangalore, India). RNA QC was confirmed by Bioanalyzer. The RNA library was prepared as per NEBNext Ultra directional RNA library prep kit. Illumina HiSeq paired end sequencing was performed.

The quality of RNA-Seq reads in the Fastq files of each sample was checked using the FastQC program (v.0.11.4) [79]. The quality of raw reads was measured using quality scores (Phred scores), GC content, per base N content, sequence length distributions, duplication levels, overrepresented sequences, and K-mer content as parameters. Trimmomatic (v.0.36) was used to remove adaptors, and low-quality sequences to rid the raw reads of any artefacts [80]. After filtering, the paired-end reads from each sample were mapped to the reference genome of *Drosophila melanogaster* (Dmel_Release_6) using HISAT2 program (v.2.0.5) [81,82]. The index for the reference genome required by HISAT2 to identify the genomic positions of each read was provided by downloading the prebuilt index for *D. melanogaster* from the HISAT2 site (http://ccb.jhu.edu/software/hisat2/manual.shtml).

Transcript assembly and relative abundances of isoforms were determined using StringTie (v.2.1.1) [83]. The merged transcripts were fed back into StringTie to re-estimate the transcript abundances using the merged structures. The read counts from this were normalized against gene length to obtain FPKM (Fragment Per Kilobase of exon model per million mapped reads) values using Ballgown [84] and TPM (Transcript Per Million) values were obtained. All FPK values in a particular sample were added and divided by 1,000,000 to get a "per million" scaling factor. Each FPK value was then divided by the "per million" scaling factor to obtain the Transcript Per Million (TPM) values. Genes and transcripts that were differentially expressed between the two genotypes were determined using DESeq2 (V1.26.0) [85]. The resulting P values were adjusted using Benjamini and Hochberg's statistical test to control the false discovery rate (FDR), and the volcano plot was generated. Bioinformatic analysis was performed by DeepSeeq Bioinformatics, Bengaluru, India.

## Y-maze olfaction assay

A Y-maze was used to carry out olfaction assays (https://www.jove.com/v/20142). In brief, the Y connector was fitted on one side with a straight tip and two tapered tips were placed on the other two connecting arms. Three vials to house flies were connected to the straight and tapered tips. Between 10–15 female flies were starved for 2–3 hours in empty vials containing filter papers soaked with water. 40 μl of the volatile to be tested was spotted onto filter paper

and loaded in one of the arms with the tapered tips. In the other arm, solvent alone was loaded. The volatiles tested were ammonia, acetophenone, and ß-ionone at concentrations indicated in the Figures. Ammonia and acetophenone were dissolved in water, while ß-ionone was dissolved in 7% ethanol. Following starvation, the flies were cold anesthetized and loaded into the tube with a straight tip. The Y-maze with flies was kept in the dark at 25˚C for 24 hours. After 24 hours, the number of flies present in each arm was counted, and the olfactory index was calculated. The olfactory index percentage = (flies in arm containing β-ionone-flies in arm containing solvent/total number of flies) x 100.

### Desiccation survival analysis

For desiccation survival analysis vials were made by adding 4.5 g of Drierite to 50-ml glass vials. Foam stoppers were then placed to cover the Drierite and prevent direct access of flies to Drierite [86]. 20 female flies were anesthetized with $CO_2$ and placed in the vials, which were sealed with Parafilm. The number of dead flies were counted at regular intervals until all the flies were dead. The experiment was repeated at least 3 times.

### Expression and purification of recombinant proteins

RNA was isolated from one day old mouse pup brains and subjected to first strand DNA synthesis using reverse transcriptase (Fermentas). The mouse MPPED2 full length coding region was amplified from the cDNA by PCR using mouse MPPED2 fwd *NcoI* and mouse MPPED2 rvs *XhoI* (5′ TGCTCGAGTTTRTAGACYKTCCCTCACATTCCAA 3′). The PCR product ($\sim$956 bp) was digested with *NcoI* and *XhoI* and ligated into *NcoI* and *XhoI* digested pPRO-ExHTC vector (Invitrogen Life Technologies, USA), and the clone verified by sequencing (Macrogen, South Korea). Wildtype mouse MPPED2 full length protein was expressed and purified as described earlier [9].

### Preparation of homogenates and RNA from mouse tissues

Brains from day 1 old pups were homogenized in homogenization buffer (50 mM Tris-Cl (pH 7.5), 2 mM EDTA, 1mM DTT, 100 mM sodium chloride, and 1x Roche protease inhibitor cocktail) using a tissue homogenizer. Samples were centrifuged at 17,000g for 30 min at 4˚C to remove insoluble material. Aliquots were made and stored at -70˚C. Total protein concentration was determined using Bradford's Method [87]. Recombinant protein (20 ng) or protein from homogenates (50 μg) were subjected to SDS gel electrophoresis and western blot analysis using a monoclonal antibody generated in the laboratory earlier [9].

RNA was prepared from the cortex of adult mice as described for the preparation of RNA from flies.

### Primary neuronal culture

For primary neuronal cultures, glial feeder layers were prepared 15 days before the culture. 4 day old mouse pups were decapitated and cerebral cortices were removed and kept in Calcium-Magnesium Free phosphate-buffered saline (CMF-PBS). Meninges were removed with the help of fine forceps to minimize non-glial cells contamination. Cortices were minced into small pieces in CMF-PBS with the help of scissors and allowed to settle down. CMF-PBS was removed and tissue was re-suspended in Trypsin-DNase solution and incubated for 5 min at 37˚C. Trypsin-DNase solution was removed and tissue was re-suspended in CMF-PBS containing DNase and incubated for 5 min at 37˚C. PBS-DNase solution was removed and tissue was gently triturated in CMF-PBS and then centrifuged at 2000 rpm for 5 min. The pellet of

tissue was suspended and gently triturated in serum containing media. This was followed by centrifugation at 2000 rpm for 5 min and the media was removed. Serum containing media (Basal Medium Eagle's, 5% fetal bovine serum, 10% horse serum, 30% glucose, 1X Pen-Strep and 1X glutamax) was added and trituration performed to obtain a single cell suspension. The suspension was plated on dishes as required. Glial cells were grown to confluency and fed at regular intervals with serum free media for 10 hours. The conditioned media was used for maintaining the primary neuronal cultures.

One day old mouse pups were decapitated and the cortex dissected, minced into small pieces in CMF-PBS and allowed to settle. Tissue was processed as described above for glial cells except that trituration was performed in neuronal plating media (Basal Medium Eagle's, 5% fetal bovine serum, 10% horse serum, 30% glucose, 1x Pen-Strep and 1x Glutamax). Re-suspended tissue was then centrifuged at 2000 rpm for 5 min and media was removed. Trituration was repeated to generate a single cell suspension. The single cell suspension was plated on coverslips in neuronal plating media, and 10 hours later, when $\sim$ 90% of the cells had adhered to the coverslip, media was replaced with serum free glial conditioned media. The serum free glial conditioned media was replaced every third day. To stop proliferation of glial cells, 5 µM of Ara-C was added to the media from day 4 onwards till the cultures were harvested.

## Immunofluorescence imaging

Cells grown on coverslip were fixed with the addition of 4% paraformaldehyde (PFA) at room temperature for 20 minutes. Cells were rinsed with PBS and permeabilized by the addition of 0.1% Triton-X100 for 10 minutes at room temperature. Blocking solution (5% BSA in PBS) was added for 1 hour at room temperature. Primary antibody (either anti MPPED2-2B1 mAb (1 µg/ml) or anti-Tuj1 (1:2500, Covance) were diluted to the desired concentration, added to the cells and incubation continued overnight at 4˚C. Cells were washed thrice with PBS, and incubated with Alexa Flour-conjugated secondary antibodies (1:200, Invitrogen) for 1 hour at room temperature. Cells were washed thrice with PBS, nuclei stained with Hoechst dye and mounted with anti-fade on glass slides. Coverslips were sealed with nail polish, and confocal imaging performed with a Zeiss LSM 880 confocal microscope with Airyscan detector. Normal mouse IgG, normal rabbit IgG and normal sheep IgG (Sigma) were used as non-specific staining controls.

## Data analysis

All data was analyzed using GraphPad Prism 9. Statistical analysis was performed across groups of flies (20–50 in each group). If two groups were present, a parametric, unpaired t-test was used to compare the groups. If more than two groups were present, one-way ANOVA was used and corrected for multiple comparisons using Tukey's test. Other details are indicated in the Legends and p values are shown in the graphs or mentioned in the Legends.

## Supporting information

**S1 Table. List of primers used in this study.**
(DOCX)

**S2 Table. List of flies used in this study.**
(DOCX)

**S1 Fig. Biochemical Characterization of dMPPED.**
(PDF)

**S2 Fig. Expression of dMPPED.**
(PDF)

**S3 Fig. Generation of *dMPPED^{KO}* flies.**
(PDF)

**S4 Fig. Full blot for Fig 3A.**
(PDF)

**S5 Fig. Odorant-binding protein expression in *dMPPED^{KO}* flies.**
(PDF)

**S6 Fig. Full blots for Fig 5B and 5E.**
(PDF)

**S1 Movie. 3D renderings of sections of a female brain from *pBAC(IT.GAL4)CG16717/ UAS-mCD8GFP* flies.**
(AVI)

## Acknowledgments

We thank Dr. Raghu Padinjat, National Centre for Biological Sciences (NCBS), for advising on genetic analysis in the initial stages of the study. We also acknowledge the efforts of Dr. Richa Tyagi and Dr. Upendra Nongthomba in early work. Microinjection of flies was performed in the Fly Facility at NCBS.

## Author Contributions

**Conceptualization:** Sveta Chakrabarti, Sandhya S. Visweswariah.

**Data curation:** Kriti Gupta, Sveta Chakrabarti, Vishnu Janardan.

**Formal analysis:** Kriti Gupta, Sveta Chakrabarti, Vishnu Janardan, Sandhya S. Visweswariah.

**Funding acquisition:** Sandhya S. Visweswariah.

**Investigation:** Kriti Gupta, Sveta Chakrabarti, Vishnu Janardan, Nishita Gogia.

**Methodology:** Kriti Gupta, Sveta Chakrabarti, Vishnu Janardan, Sanghita Banerjee, Swarna Srinivas, Deepthi Mahishi.

**Project administration:** Sandhya S. Visweswariah.

**Resources:** Sandhya S. Visweswariah.

**Software:** Sanghita Banerjee.

**Supervision:** Sandhya S. Visweswariah.

**Validation:** Sandhya S. Visweswariah.

**Writing – original draft:** Sandhya S. Visweswariah.

**Writing – review & editing:** Kriti Gupta, Sveta Chakrabarti, Sandhya S. Visweswariah.

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
