## [Decision Letter · Decision Letter 0]

14 Apr 2023

Dear Dr Visweswariah,

Thank you very much for submitting your Research Article entitled 'Neuronal expression in Drosophila of an evolutionarily conserved metallophosphodiesterase reveals pleiotropic roles in longevity and odorant response.' to PLOS Genetics.

The manuscript was fully evaluated at the editorial level and by independent peer reviewers. The reviewers appreciated the attention to an important problem, but raised some substantial concerns about the current manuscript. Based on the reviews, we will not be able to accept this version of the manuscript, but we would be willing to review a much-revised version. We cannot, of course, promise publication at that time.

If you decide to revise the manuscript for further consideration at PLOS Genetics, please aim to resubmit within the next 60 days, unless it will take extra time to address the concerns of the reviewers, in which case we would appreciate an expected resubmission date by email to plosgenetics@plos.org.

We are sorry that we cannot be more positive about your manuscript at this stage. Please do not hesitate to contact us if you have any concerns or questions.

Yours sincerely,

Pablo Wappner

Academic Editor

PLOS Genetics

Gregory Barsh

Editor-in-Chief

PLOS Genetics

Reviewer's Responses to Questions

**Comments to the Authors:**

Reviewer #1: Is the study “Neuronal expression in Drosophila of an evolutionarily conserved metallophosphodiesterase reveals pleiotropic roles in longevity and odorant response”, Gupta K. and collaborators characterized for the first time the multiple roles of an ortholog of the mammalian metallophosphodiesterase MPPED1/2 called dMpped. They demonstrate that dMpped can hydrolyze cAMP and cGMP, and that dMpped is expressed in development but, like in mammals, is also highly enriched in the adult brain. By performing RNA-seq analysis on dMpped KO flies, the authors showed an up-regulation of genes associated with the bacterial defence pathway associated to a shorter lifespan of these flies suggesting that a certain level of immune response or inflammation is detrimental to lifespan thus due to an increased cAMP/cGMP. The RNA-seq analysis also revealed that dMpped KO flies exhibit higher expression of odorant-binding protein together with an impaired olfactory acuity to some odours. Thus, showing a diverse role for dMpped. They demonstrated that all phenotypes could be rescued by expression of dMpped in the brain. And finally, expressing the mammalian MPPED2 in neurons also partially rescued the shortened lifespan observed in dMpped KO flies, nicely demonstrating a conserved role from Drosophila to mammals.

The study is well organized, easy to read and to understand. Overall, it is a high quality piece of work and constitutes a significant advance to the field. Below are some suggestions to the authors to improve the manuscript.

Major Comments:

1) The presentation of the statistical analysis is very confusing in the legends. I would very much recommend a detailed description of the statistics in the Material and Methods. For example, we never know if the statistics are performed on the groups or on the individuals. In Figure 4C, the legend is confusing. I assume from the figure that each dot corresponds to an experiment. Hence the legend should only cite the number of experiments, especially since the statistics are supposed to be made on the groups and not the individuals. Also, legends indicate the use of ANOVA for statistics, for the high concentration experiment, you cannot perform ANOVA since there are only 2 groups.

2) The olfaction acuity experiment at low concentration seems to have twice more data points than at the low concentration (Figure 4C). It would be appropriate to increase the data points of the high concentration experiments to at least 6. Similarly, in Supplemental Figure 4B-C, 3 data points for an olfactory test is very low and the number of experiments ought to be increased.

3) A section describing the survival assay is missing in the Material and methods. Only survival to desiccation is present.

4) The authors argue that the expression of dMpedd in possibly very specific neurons alters odorant perception and hygrosensation. Elav-Gal4 is a general driver targeting all neural populations. This argument would be greatly improved if the authors could replicate the odorant perception experiments and the resistance to desiccation in a more specific subset of neurons. For example, by using orco-Gal4 which targets the neural populations of the antennal lobes. Or by using GH-146-Gal4 which targets the OPNs, particularly since these neurons show a high dMpedd expression (Figure 4A).

Minor Comments:

1) Line 79: hydrolyze instead of « hydroylze »

2) Line 93: hydrolyze instead of hdyrolyze

3) Line 131: After “We noted high expression in the brain“. Figure 1E should be cited, not 1D.

4) Line 138: Figure 1F should be cited, not 1E

5) Line 194: The authors forgot to cite Figure 3C

6) Line 198: Supplemental Figure 2D does not show that dMpped can hydrolyze cAMP and cGMP. The right citation is “Supplemental Figure 1D”

7) The olfactory acuity to ß-ionone has been tested at “high” and “low” concentration, and a significant effect was observed only at low concentration (Figure 4C). For the experiment using ammonia and acetophenone, only one concentration has been used. Please describe whether these concentrations should be considered as “low” or “high” and why is there a difference in results depending on the concentration.

8) Figure 1D-E: Add a title saying dMpped expression.

9) Figure 1F: Add the genotype.

10) The expression pattern data from Figure 1F is a very interesting data, providing a movie of the full scan would be very useful to the community. It would for example help to identify which antennal lobe glomeruli are stained and thus give a more precise idea of dMpped neural expression.

11) Supplemental Figure 2B: The B is missing in the figure

12) Figure 2D: I agree that the data shows a “rescue” but you need to show that dMppedKO and Dredd; dMppedKO are significantly different. It would also help the reader to understand that there is a difference by showing statistics in the figure.

13) Figure 3B: Same comment as Figure 2D: I agree that the data shows a “rescue” but you need to show that dMppedKO and elav-Gal4>dMpped; dMppedKO are significantly different. It would also help the reader to understand that there is a difference by showing statistics in the figure.

14) Figure 4A: specify in the title that you are looking at dMpped expression.

15) Figure 4: The titles “A” and “B” are not at the same height, please realign.

16) Figure 4B: The titles “Obp28” and “Obp59” are not at the same height, please realign.

17) Figure 4D: Same comment as Figure 2D and 3B: I agree that the data shows a “recue” but you need to show that dMppedKO and elav-Gal4>dMpped; dMppedKO are significantly different. It would also help the reader to understand that there is a difference by showing statistics on the figure.

18) Figure 5A: The legend says “Right panel: survival curves of indicated fly lines.” This should be removed, there is no survival curve in Figure 5A.

19) Figure 5E: Same comment as Figure 2D, 3B and 4D: I agree that the data shows a “recue” but you need to show that dMppedKO and elav-Gal4> Mpped2; dMppedKO are significantly different. It would also help the reader to understand that there is a difference by showing statistics in the figure.

20) The authors show that obp expression and so olfaction has an impact on lifespan and suggest a connection with anti-microbial peptide and crystal cell production. It is known that olfaction has a strong impact on hematopoiesis (Shim et al., Cell 2013), could the authors discuss their results in line with this publication?

21) In Figure 4A, the authors analyzed the expression of dMpped in 3- and 30-days old flies. Even though they mention in the discussion that dMpped plays a role in aging (lines 281-282), they never discussed the differences of expression between young and old flies. For example, it seems that there is a strong increase of dMpped expression in peptidergic neurons which are involved in many processes such as metabolism or even regulation of olfactory sensitivity. I think it would provide useful information to discuss such interesting results.

22) Virgin females have been used for experiments (line 394, 455). Do the authors know if the mating status of female flies’ influences dMpped levels? Also, are the 35 days old flies used for RNA-sequencing virgin or mated (Figure 4A)? If not, can the authors discuss that the differential expression between 3days virgin female and 35 days old mated females is due to aging and not mating status?

23) The story is very interesting, and so I think it would help the reader if the authors could provide a schematic of the dMpedd pathway, showing the substrates, reactions, as well as the diverse roles investigated. This could be used as a final figure summarizing the story.

Reviewer #2: This work explores the participation of the fly ortholog of the mammalian genes MPPED1/MPPED2, a metallophosphoesterase named dMpped, in longevity, immune response, and physiology. The work is conceptually interesting and brings novel insight into the function of MPPED proteins. The last part characterizing neural expression in mammalian neuros, is a good complement to the work. Overall the work is suitable for PLOS GENETICS, but some revision needs to be performed before is ready for publication.

Major comments

Figure 1, panel F (brain expression), although localization of the label makes sense with what the authors indicate, I recommend using higher magnification images and counterstaining with a neuropil marker to visualize the structures more clearly. Additionally, in the female brain, it seems to be an expression in the optic lobe (in the male brain, there is no labeling in this region, but lamina is absent). Is this protein expressed in photoreceptors or lamina neurons in the visual system? If that is the case, please include this in the description, accompanied by higher magnification images. It is very hard to see localization in the antennal lobe with this magnification and without a proper counterstaining

Line 205: Authors propose that neuronally elevated cGMP in dMppedKO flies contributes to the life span regulation in the fly, is there any previous work indicating that cGMP concentration in neurons is linked to lifespan?

Figure 3: What explains the lifespan differences for control flies between the chart in Figure 2D and 3B? I imagine the sex of the fly, but this should be indicated in the figure legend

The specific requirement of neuronal expression of Mpped is intriguing, is this connected with the immune effect, and is this independent of the function in the regulation of the expression of Obps?

Line 218, the authors indicate that deletion of Obp28 results in a reduced preference for b-ionone. Flies KO for dMpped have increased levels of Obp28, but they also present reduced preference for this molecule. Can the authors elaborate on this observation? Could it be that pleiotropic effects in dMpped mutants somehow mask an expected increase in attraction by b-ionine, given the increase in Obp28 expression in the mutant? It is hard to make conclusions without rescue experiments reducing the expression of Obps in the dMpped KO genetic background, using, for instance a mutation in heterozygosity or a RNAi

Line 236, similarly the authors observe that Obp59a is upregulated in dMpped KO flies. However, the phenotype of the dMpped mutant is the same as the mutant of the Obp59. I would expect something else. Again, a genetic interaction experiment could show that reducing Obp59 directly in the dMpped KO genetic background could modify the phenotype observed

Minor comments

In lines 128 and 133, the authors state that dMpped2 (not dMpped) is expressed in various stages, this refers to a previous name? please explain

Figure 1: Please add a schematic indicating the structure of the dMpped gene with the GAL4 insertion, is the GAL4 in the same direction as the gene?

Line 300, The authors indicate in their discussion that elevated levels of cAMP and cGMP in dMpped flies could lead to activation of IMD in the fly brain, are there any evidence on the effect of elevated cAMP/cGMP in neurons, and its relation with life span?

Line 312, the authors indicate that a relation between Obp and immunity has been shown outside the nervous system, but how Obp expression in neurons could be related to IMD and life span?

Line 382, is the temperature of the reaction correct? (or is 37C)

Reviewer #3: Gupta et al present the first biochemical and genetic analysis of the conserved Drosophila metallophosphodiesterase dMpped. They find the dMpped enzyme shares with its human homologs the ability to hydrolyze cyclic nucleotides, including derivatives of the second messengers cAMP and cGMP, and use an enhancer trap to detect enhanced dMpped expression in a few tissues, including the antennal lobes and CNS. KO animals have reduced lifespans and RNA-seq of whole animals detect changes in antimicrobial peptides (AMPs) and odorant receptors (ORs). Re-expression of dMpped in neurons of null animals rescues lifespan and AMP production, as does a hypomorph of Dredd, an IMD component that is required for Relish-driven expression of AMP genes. The authors interpret this data to suggest that elevated innate immunity is a driver of shortened lifespan in dMpped mutants. The second section of the data focus on two overexpressed odorant receptors with known odor specificities and proceed to discover that dMpped loss is associated with reduced sensitivity to the corresponding odorant.

Overall, the study presents a useful new genetic model of deficiency for the sole Mpped protein in flies that could bypass the experimental challenges of multiple Mpped homologs in mammals. The data are well controlled and document clear phenotypes that can be mapped to specific cells using Gal4s. The unique roles of dMpped in both longevity and odorant response shed light on the roles of an uncharacterized enzyme, but also have implications in mammalian nervous system dysfunction. However, as the authors’ address in the Discussion, there is a notable mechanistic gap between altered cyclic nucleotide metabolism in neurons and changes in the RNA-seq, e.g., AMPs and ORs expression, and which cells these changes actually occur in. Some level of insight into this issue would help strengthen the the study.

Major points:

1. The link between cyclic nucleotides and AMP production in dMpped mutants is very unclear and could involve systemic signals between different cell types. AMPs are normally made in the major immune organ, the fat body. Does Mpped loss in neurons lead to cell autonomous production of AMPs by adult brain neurons – which would be unprecedented as far as I am aware – or are the AMP transcripts in the RNA-seq coming from the fragment of fat body located in the head? This info would have a big impact on models of dMpped roles in neurons etc.

2. In addition, the authors seem to presume that dMpped loss elevates AMPs in the absence of infection (lines 160-161). However, all flies have chronic low-level microbial infections. If they want to claim that dMpped loss is sufficient for AMP expression, they should rear these flies in axenic conditions.

3. There is little to no explanation as to why an increase in Obp 28 and Obp59 levels in a dMpped ko mimic those seen in Obp28/Obp59 knockout/deleted flies. What is the explanation for this paradoxical relationship between Obp levels and odorant response? Could the upregulation of these OR mRNAs be an indirect compensatory effect of insufficient cyclic nucleotides to act as second messengers following odorant binding? At a minimum, this paradox needs to be addressed in the text.

4. There seem to be distinct differences between the male and female brain staining for dMpped in Figure 1F, but no mention of this within the text. This data suggest possible sex-specific roles for dMpped. This issue is significant, as most experiments use females, or virgin females only, with no mention for the underlying reasons for this decision.

5. The experiments are missing controls for Dredd hypomorphs alone and elav>MPPED2 alone (without dMppedKO in the background). We need to know that rescue is not solely due to the expression of a Dredd hypomorph or MPPED2.

6. Why is the purified protein in Figure 5B running at a different size than the Mpped blotted from the mouse brain? This is not explained in the text and the blot is poorly labelled. Is there a non-specific band present in the lanes from the brain samples?

Other comments

7. The argument that AMP overexpression contributes to reduced lifespan in dMppedKO flies could be strengthened by performing RT-qPCR to assess the levels of the other AMPs mentioned (Drs and Dro) in Figure 2. Considering that RT-qPCR for all AMPs mentioned (Dpt, Dro, and Drs) was performed for conclusions made in Figure 3.

8. There is inconsistency in the use of dMpped, CG16717, dMPPED, and dMpped2 within the text and figures and figure legends

o See line 128,133, Supplemental Figure 2C, Supplemental Figure 2E, Supplemental Figure 4 etc…

9. Please highlight H51 and D49 in Figure 1A

10. Separate the inset and the graph into different panels in Fig 1B

11. Missing figure reference in line 113.

12. Missing reference in line 132.

13. No explanation of why flies were backcrossed for 9 generations for experiments (line 184)

14. Define PKG in line 205.

15. Format Figure 1 D and E in same manner as other qPCR graphs in the paper

16. Lines 123-124 reference little detectable activity of MPPED2 with Ni2+ and reference data in Supplemental figure 1A, however, these data do not appear in this figure.

17. The blot in Figure 3A for dMpped is not the same blot as in supplemental figure 3, although it is stated that they are the same in the figure legend.

18. Missing reference in sentence on line 201.

**Have all data underlying the figures and results presented in the manuscript been provided?**

Reviewer #1: Yes

Reviewer #2: Yes

Reviewer #3: Yes

PLOS authors have the option to publish the peer review history of their article (what does this mean?). If published, this will include your full peer review and any attached files.

Reviewer #1: **Yes: **Dr. Guy Tanentzapf, The University of British Columbia

Reviewer #2: No

Reviewer #3: No

---

## [Decision Letter · Decision Letter 1]

29 Jul 2023

Dear Dr Visweswariah,

Thank you very much for submitting the revised version of your Research Article entitled 'Neuronal expression in Drosophila of an evolutionarily conserved metallophosphodiesterase reveals pleiotropic roles in longevity and odorant response.' to PLOS Genetics.

The manuscript was evaluated at the editorial level, as well as by the same three reviewers that evaluated the original manuscript. The reviewers generally feel that the manuscript is very much improved, although one of them (reviewer #2) still has some concerns that we ask you address in a new revised version manuscript. As you will see below, reviewer #3 only suggests very minor modifications in the text.

We therefore ask you to modify the manuscript according to the reviewers' recommendations. Your revisions should address the specific points made by each reviewer.

Yours sincerely,

Pablo Wappner

Academic Editor

PLOS Genetics

Gregory Barsh

Editor-in-Chief

PLOS Genetics

Reviewer's Responses to Questions

**Comments to the Authors:**

Reviewer #1: The authors did a very nice job addressing all the points raised in the first round of review.

Reviewer #2: Many of my previous comments have been reasonably addressed in this second version of the manuscript. However, I still feel that the description of the GAL4 line that reports dmpped expression lacks good characterization, the movies provided do not have enough quality, and there is no neuropil counterstaining (e.g. NCadherin, Dlg, Brp, or other) to observe the antennal lobe. The images provided in the supplementary do not allow to precisely define that all the projections are olfactory, and some seem to be gustatory. Furthermore, the female brain, used as an example, keeps part of the lamina in the optic lobe, unlike the male brain. Therefore, it is incorrect to conclude that there is a differential expression in the male vs. female optic lobe because the male does not have part of the brain. Again here, a NCadherin staining would help. The GH 146-GAL4 is expressed in projection neurons in the antennal lobe, among other populations, the authors need to define best if dmpped is expressed only in olfactory neurons or also in projection neurons.

Reviewer #3: Thank you for your efficient efforts in responding to my critiques, and also to those of the other two reviewers. There is an impressive amount of new data in the revision.

My final comment/request is that the authors reformat the lifespan data in Figs 2D and 3B. The current format for both undersells the data that is described in the legends. The 2D and 3B have single lines that make it appear that the experiment was done once. Importantly, the figure legends for both panels describe multiple replicates for each genotype, so presumably the single colored lines shown for each genotype is either an average or one representative "run". The data would be more authentic and rigorous if the graphs were shown with error bars for each timepoint, together with a p-value comparing the avg lifespan across all three replicates of each genotype.

**Have all data underlying the figures and results presented in the manuscript been provided?**

Reviewer #1: Yes

Reviewer #2: Yes

Reviewer #3: Yes

PLOS authors have the option to publish the peer review history of their article (what does this mean?). If published, this will include your full peer review and any attached files.

Reviewer #1: **Yes: **Guy Tanentzapf

Reviewer #2: No

Reviewer #3: No

---

## [Decision Letter · Decision Letter 2]

16 Aug 2023

Dear Dr Visweswariah,

Thank you very much for submitting your Research Article entitled 'Neuronal expression in Drosophila of an evolutionarily conserved metallophosphodiesterase reveals pleiotropic roles in longevity and odorant response.' to PLOS Genetics.

The revised version of your manuscript was evaluated at the editorial level and by one of the reviewers that evaluated the previous version. The reviewer found that the manuscript has improved a but identified some concerns that you need to address in a revised manuscript.

We therefore ask you to modify the manuscript according to the review recommendations. Your revisions should address the specific points made by the reviewer.

Yours sincerely,

Pablo Wappner

Academic Editor

PLOS Genetics

Gregory Barsh

Editor-in-Chief

PLOS Genetics

Reviewer's Responses to Questions

**Comments to the Authors:**

Reviewer #2: Unfortunately, one my most important comments has not been addressed in this new version. It is clear to me that the authors are presenting data in which the female brain keeps the lamina neuropil (the most lateral neuropil in the optic lobe), which has expression of the driver, while this neuropil is absent in the male brain presented. It is also evident from the Dapi staining that lamina is absent only in males and from the stacks in the supplementary data. Commonly, the lamina is lost during brain dissections for whole mount staining. The reference cited (Rein et al., 1999) indeed describes differences between males and females, but both sexes have the same neuropils, they only differ in size. If the authors want to make the case that male and female have differences in expression of the driver in the optic lobe, they need to present preparations in which brains of both sexes are in the same condition.

**Have all data underlying the figures and results presented in the manuscript been provided?**

Reviewer #2: Yes

PLOS authors have the option to publish the peer review history of their article (what does this mean?). If published, this will include your full peer review and any attached files.

Reviewer #2: No

---

## [Decision Letter · Decision Letter 3]

7 Sep 2023

Dear Dr Visweswariah,

We are pleased to inform you that your manuscript entitled "Neuronal expression in Drosophila of an evolutionarily conserved metallophosphodiesterase reveals pleiotropic roles in longevity and odorant response." has been editorially accepted for publication in PLOS Genetics. Congratulations!

Yours sincerely,

Pablo Wappner

Academic Editor

PLOS Genetics

Gregory Barsh

Editor-in-Chief

PLOS Genetics

Comments from the reviewers (if applicable):

Reviewer's Responses to Questions

**Comments to the Authors:**

Reviewer #2: My comments have been addressed satisfactorily. I congratulate the authors for this important contribution

**Have all data underlying the figures and results presented in the manuscript been provided?**

Reviewer #2: Yes

PLOS authors have the option to publish the peer review history of their article (what does this mean?). If published, this will include your full peer review and any attached files.

Reviewer #2: No

**Data Deposition**

http://datadryad.org/submit?journalID=pgenetics&manu=PGENETICS-D-23-00128R3

**Press Queries**

---

## [Editor Report · Acceptance letter]

18 Sep 2023

PGENETICS-D-23-00128R3 

Neuronal expression in Drosophila of an evolutionarily conserved metallophosphodiesterase reveals pleiotropic roles in longevity and odorant response. 

Dear Dr Visweswariah, 

We are pleased to inform you that your manuscript entitled "Neuronal expression in Drosophila of an evolutionarily conserved metallophosphodiesterase reveals pleiotropic roles in longevity and odorant response." has been formally accepted for publication in PLOS Genetics! Your manuscript is now with our production department and you will be notified of the publication date in due course.

With kind regards,

Anita Estes

PLOS Genetics

On behalf of:
